# Studies toward the Use of Ionic Liquids and Supercritical CO₂ for the Recovery and Separation of Praseodymium from Waste Streams

Rene Rodriguez [1,*], Donna Baek [2], Mary Case [2] and Robert Fox [2]

1 Department of Chemistry, Idaho State University, Pocatello, ID 83209, USA
2 Idaho National Lab, Idaho Falls, ID 83415, USA; donna.baek@inl.gov (D.B.); mary.case@inl.gov (M.C.); robert.fox@inl.gov (R.F.)
* Correspondence: rodrrene@isu.edu; Tel.: +1-208-282-2613

**Abstract:** Waste streams from the incineration of metal-containing materials like such as computer processor boards and batteries may contain critical rare earth elements like praseodymium. Data on the solubility of Pr compounds and on their distribution coefficients in supercritical $CO_2$/ionic liquid two-phase systems are important to determine if an ionic liquid/supercritical $CO_2$ two-phase approach is feasible toward the recovery of a particular metal. This work provides data on the solubility of various praseodymium compounds in butyl-methyl-pyrrolidinium bis(trifluoromethylsulfonyl)imide (BMPyTf2N) ionic liquid and on the distribution coefficients of these praseodymium compounds in the supercritical $CO_2$ phase of the two-phase BMPyTf2N ionic liquid/supercritical $CO_2$ system, with and without a tributyl phosphate additive.

**Keywords:** praseodymium recovery; ionic liquid; supercritical $CO_2$; two-phase system; electrochemical reduction





## 1. Introduction

In recent years, there has been increasing interest and effort in recovering rare-earth metals (REMs) from discarded computer hardware, batteries, lighting, and other discarded items. Some or all of the following steps may be required for metal recovery: (1) dissolving the waste material in an acid solution to oxidize the metal and bring it into solution as a nitrate, sulfate, or another salt, (2) separating the metal salt from the rest of the mixture, (3) converting the separated metal salt into a form that is more amenable to final processing, (4) final processing of the converted REM salt to transform it into the preferred recovered form.

Often the preferred and functional final form for the recovered REM is the elemental, zero-valent metal. Since metals are often in a mixed oxidized form due to the recovery methodology, separation and electrochemical reduction of REMs are necessary in the recovery process. Chemical substances that have shown promise for the electrochemical reduction of REMs are ionic liquids (ILs) [1–5]. Several types of ILs and metal salts have been studied to determine their overall potential for the electrodeposition of pure REMs. Most studies report that at least some zero-valent metal is deposited on the cathode, but both the oxidation state and the purity of the deposited metal are often not optimal, and the water content likely plays a significant role in the deposition [5,6].

It has become apparent that once the separation of the metal from the waste stream has been accomplished, the three key steps in recovering REMs as neutral metals, through electrochemical reduction in ILs, are: (1) choosing an ionic liquid (IL) solution that provides the most appropriate electrochemical medium for electrodeposition of REM, (2) converting the recovered REE salt into an ionic compound that is more amenable to electrochemical reduction by the IL, and (3) minimizing the co-deposition of degradation products from the electrochemical medium.

Several factors must be considered in choosing an appropriate IL solution and the REE compound that is amenable to reduction in the IL. The electrochemical window of the IL solution must be large enough to encompass the potentials needed for metal reduction at the cathode, the ions must have good mobility and conductivity in the IL solution, and the anions/ligands surrounding the metals should have good stability and minimally interfere with the deposition process. The electrochemical windows of many ILs have been measured and tabulated, and it has been reported they are somewhat sensitive to the electrodes used. There is also no standard cut-off current density in use for the measurement [7,8]. The identity of both the cation and the anion will affect the electrochemical window, and the anion oxidation stability and the cation reductive window largely determine the anodic and cathodic limits for the IL. Phosphonium cations have been found to have perhaps the largest reductive windows, and limits for some ILs follow the sequence trihexyl-tetradecylphosphonium ($P^+_{6,6,6,14}$) > 1-butyl,1-methylpyrrolidinium ($BMPyr^+$) > 1-ethyl 3-methyimidazolium ($EMI^+$) > 1-(2-methoxyethyl) 1-methylpiperidinium ($MOEMPip^+$ > n-butyl 3-methylpyridinium ($BMPy^+$). Additionally, anion stability was reported to follow the sequence bistrifluoromethysulfonylimide ($Tf2N^-$) > trispentafluorotrifluorophosphate ($TPTP^-$) > trifluoromethanesulfonate (triflate) (Tf) > dicyanamide ($DCA^-$) > trifluoroacetate ($TFA^-$) [7].

Reasonably high mobility and conductivity of the electrochemical medium are important for an efficient deposition that is not limited by the movement of REM cations to the cathode. Ionic liquids with long and/or bulky alkyl tails have relatively high viscosity and low conductivity. Phosphonium cations like $P^+_{6,6,6,14}$ have a large electrochemical window but are relatively viscous. Pyrrolidinium cations, like $BMPyr^+$, have a large window as well, and the viscosity of the IL BMPyrTf2N is about a fourth of that of $P_{6,6,6,14}$Tf2N [9,10]. The large electrochemical window, relatively moderate viscosity, touted stability, and hydrophobicity of the $Tf2N^-$ anion suggest that BMPyrTf2N is a good IL for electroreduction, and several researchers have investigated it for the purpose of electrodepositing various REMs including Dy, Ce, Eu, Sm, Nd, and Pr [4,11,12].

As pointed out by Bourbous et al. and others [3,13], the anion of a REM salt is also important for the recovery. Solubility of the various metal salts is important for a cost-effective electrodeposition process. Some metal nitrates likely are somewhat soluble in an IL like BMPyrTf2N, but nitrate is in its highest oxidation state, and oxidation at the anode would likely mean that the anion of the IL or the solvent would have to be oxidized. Halides would be good candidates, since their ions could be oxidized to their elemental state. Thus, it is important to convert oxidized REM recovered from waste into an appropriate REM salt based on a judicious consideration of two factors: (1) the solubility of the REM salt in the IL that will be used for the electrodeposition and (2) the oxidized product formed during the electrodeposition, presumably from the anion of the salt or from the solvent. The oxidized product is very important, since it may interfere with the electrodeposition of the pure metal at the cathode. The presence and detrimental effect of impurities from oxidized products in the electrochemical deposition of a pure metal have been noted by several groups investigating the electrochemical deposition of REMs. If these oxidized products could be separated from the ionic liquid during the deposition, then the depositions could yield higher-quality REM deposited at the cathode.

In principle, performing the electrodeposition in the presence of supercritical carbon dioxide, $scCO_2$, could in fact provide the desired separation of potential impurities and interferants such as the oxidized products from the ionic liquid, during the deposition process. The promising prospect of using $scCO_2$ to remove impurities during the process has been pointed out by a few researchers [14,15], but to date only a very limited number of research studies have investigated such an approach [16].

Three properties of the combination of IL and $scCO_2$ make this in situ "simultaneous metal reduction–impurity removal" possible: (1) the reasonable solubility of $scCO_2$ in ionic liquids, (2) the decreased viscosity of the IL due to the dissolved $CO_2$, (3) the complete insolubility of the IL in $scCO_2$. The insolubility of the IL in $scCO_2$ and the simultaneous

solubility of $CO_2$ in the IL is a very unusual property [17], and the net result is the existence of both an IL phase with some dissolved $CO_2$ and an $scCO_2$ phase with assumedly a negligible amount of IL. The $CO_2$ in the IL phase may solubilize the impurities and transfer them into the $scCO_2$ phase. If the electrodeposition of the REM is performed in the IL phase and the $scCO_2$ phase is maintained under a slight flow, the impurities could be removed from the area of deposition. A schematic of a possible overall approach is presented in Figure 1.

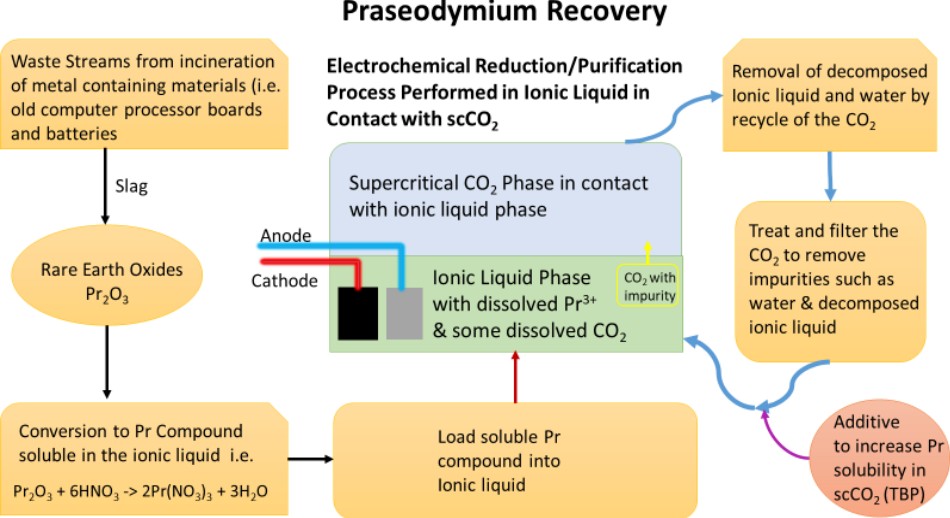

**Figure 1.** Example of Pr recovery with two-phase impurity removal.

The prospect of performing an impurity-free electrodeposition of REMs in an ionic liquid phase in contact with a $scCO_2$ phase seems very promising. However, in order to consider this type of process, additional information is needed. Information on the solubility of the various REMs, such as Praseodymium in both IL and $scCO_2$ would allow for a decision on the best REM salt to use in the deposition, assuming it could be made from the recovered oxidized REM. Ideally, the salt would have reasonable solubility in an IL with a sufficient electrochemical window and limited solubility in $scCO_2$. Additionally, an additive like tributyl phosphate (TBP) could be added to help solubilize the oxidation byproducts and the impurities and move them into the $scCO_2$ phase; the solubility of the metal compound in the $scCO_2$ with the TBP additive would also need to be known.

The purpose of this work was to further the availability of solubility measurements toward the recovery of the Pr metal through electrodeposition of Pr from its metal salt in a two-phase system containing the BMPyrTf2N ionic liquid and supercritical $CO_2$. The solubilities of several Pr compounds were determined in the IL 1-butyl-methylpyrrolidinium bis(trifluoromethysulfonyl)limide, (BMPyrTf2N),. The solubilities of selected Pr salts in just supercritical carbon dioxide and also in the $scCO_2$ phase of the BMPyrTf2N IL/$scCO_2$ two-phase system, both without and with a tributyl phosphate (TBP) cosolvent, were also determined. The amount of Pr solubilized was determined in a way similar to that used by other researchers [18], from the molar absorptivity of the $Pr^{3+}$ cation at the maximum of one peak in its absorption spectrum and then using Beer's law to determine the $Pr^{3+}$ concentration in the saturated solution and $scCO_2$ phase of the two-phase system.

## 2. Results

The praseodymium(III) salts used for the solubility studies were chloride ($PrCl_3$), carbonate ($Pr_2(CO_3)_3$), triflate ($PrTf_3$), acetate ($Pr(C_2H_3O_2)_3$), Tris(6,6,7,7,8,8,8-heptafluoro-2,2-dimethyl-3,5- octanedionato) ($Pr(fod)_3$), nitrate ($Pr(NO_3)_3$), acetylacetonate, $Pr(acac)_3$, hexafluoroacetylacetonate ($Pr(hfacac)_3$), and bis(trifluoromethylsulfonylimide) ($Pr(Tf2N)_3$). The solubilities of these Pr salts were measured at a temperature of 23 °C. In most cases, the salt solubilities were determined directly in its container and also after it was dried for

several hours in a vacuum oven. The solubilities were calculated using the Beer–Lambert law. The values calculated were based on the absorbance values at the maximum in the absorbance peak for Pr(III) at 590 nm and a molar absorptivity of 1.8 L/(mol cm). The spectra of several of the Pr salts dissolved in BMPyrTf2N IL are shown in Figure 2.

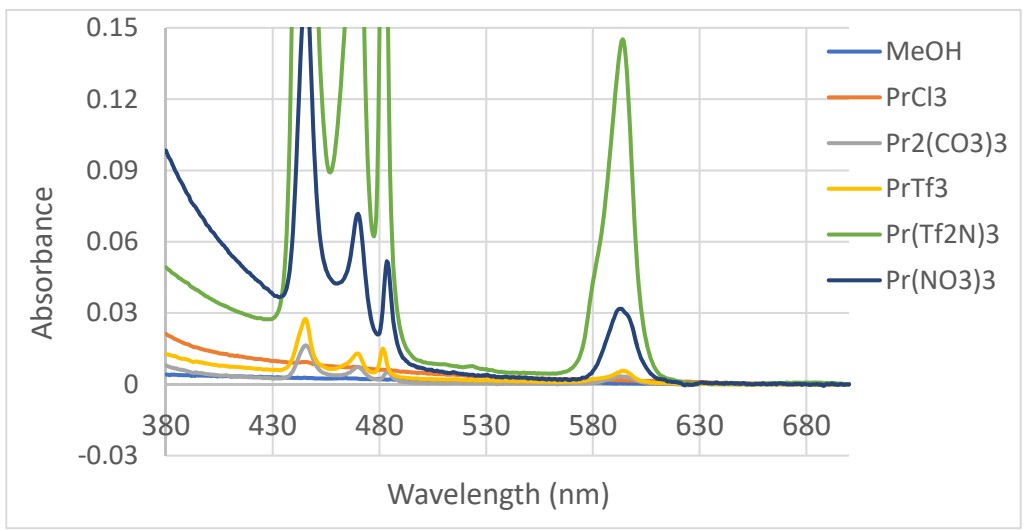

**Figure 2.** Absorbance spectra of several Pr salts in the visible region.

The Pr salt solubilities derived from the absorbance spectra are reported in $gPr/L_{soln}$ in Table 1. Multiple measurements of the solubility of dried $Pr(NO_3)_3$ salts were used to estimate the relative standard error associated with these measurements. The estimate of the error was found to be on the order of $+/-30\%$ of the measured value. Note that for all Pr salts measured, the solubility of the dried compound was significantly higher.

**Table 1.** Solubility of several Praseodymium salt compounds in 1-butyl-1-methylpyrrolidinium bis(trifluoromethylsulfonyl)imide at 25 °C.

| Praseodymium Salt | Solubility (g Pr/L) in BMPyrTf2N IL | Praseodymium Salt | Solubility (g Pr/L) in BMPyrTf2N IL |
|---|---|---|---|
| $PrCl_3$ | below detection | $Pr(acac)_3$ | below detection |
| $PrCl_{3\ (dry)}$ | 0.012 | $Pr(acac)_{3\ (dry)}$ | 0.34 |
| $Pr_2(CO_3)_{3\ (dry)}$ | 0.91 | $PrTf_{3\ (dry)}$ | 1.3 |
| $Pr(C_2H_3O_2)_3$ | below detection | $Pr(FOD)_{3\ (dry)}$ | 0.92 |
| $Pr(C_2H_3O_2)_{3\ (dry)}$ | below detection | $Pr(Tf_2N)_{3\ (dry)}$ | 35 |
| $Pr(NO_3)_3$ | 0.25 | $Pr(hfacac)_{3\ (dry)}$ | 69 |
| $Pr(NO_3)_{3\ (dry)}$ | *6.5* | - | - |

Since the electrochemical reduction and rate of electrodeposition of the zero-valent Pr metal is largely governed by the concentration of Pr in the ionic liquid electrolyte, a Pr solubility greater than 40 mM in the IL would likely be necessary for the deposition process. Only three of the Pr salts tested had a solubility in BMPyrTf2N IL greater than 40 mM. They were $Pr(NO)_3$, $Pr(Tf_2N)_3$, and $Pr(hfacac)_3$. These three Pr salts were then chosen for further investigation.

Researchers have found that $scCO_2$ is somewhat soluble in ionic liquids, but at least some ionic liquids are not soluble in $scCO_2$. Thus, when $CO_2$ is added to a vessel containing an ionic liquid and is pressurized above the critical point, two phases are present, an IL phase with some $CO_2$ dissolved in it, and a $scCO_2$ phase with presumably little to no IL present in it. If $scCO_2$ is to be useful in the electrochemical recovery of zero-valent REMs

like Pr, then it must be able to pull oxidation products and other impurities out of the IL phase and into the scCO$_2$ phase without lowering the Pr content in the ionic liquid.

Data on the solubility of metal salts in the presence of scCO$_2$ were gathered. Three types of measurements were made: (1) the first measurement determined the solubility of the three metal salts with reasonable IL solubility, i.e., Pr(NO)$_3$, Pr(Tf2N)$_3$, and Pr(hfacac)$_3$ in scCO$_2$ alone; (2) 40 mM solutions of each Pr salt dissolved in BMPyrTfN IL were treated with scCO$_2$ to obtain a two-phase system, and the solubility of Pr in the scCO$_2$ phase was measured; (3) the third solubility measurements were of the Pr present in the scCO$_2$ phase in the same two-phase system in the presence of enough TBP to form a 5 mol% solution with respect to the number of moles of scCO$_2$ present. The results from these solubility studies in neat scCO$_2$ and in the scCO$_2$ phase of the mixture with the IL are shown in Table 2, and the results for the scCO$_2$ phase of the mixture with the TBP additive are presented in Table 3. The detection limit for Pr using the absorbance method was estimated to be 0.01 g Pr/L. As can be seen in the tables, in many cases the absorbance values were below the detection limit of 0.01 gPr/L associated with this method.

**Table 2.** Results from the solubility (solub.) studies of Pr salts in neat scCO$_2$ and in the scCO$_2$ phase in contact with the IL phase containing the metal salt at 308 K and three pressures.

| Pressure (MPa) | Solub. (g Pr/L) in Pure scCO$_2$ | | | Solub. in scCO$_2$ Phase Contacting IL Phase | | |
|---|---|---|---|---|---|---|
| | PrTf$_2$N | Pr(hfacac)$_3$ | Pr(NO$_3$)$_3$ | PrTf$_2$N | Pr(hfacac)$_3$ | Pr(NO$_3$)$_3$ |
| 14 | 0.08 | 0.60 | <0.01 | <0.01 | <0.01 | <0.01 |
| 28 | 0.23 | 1.16 | <0.01 | <0.01 | <0.01 | <0.01 |
| 34 | 0.27 | 1.21 | <0.01 | <0.01 | 0.05 | <0.01 |

**Table 3.** Results from the solubility studies of Pr salts in the scCO$_2$ phase in contact with the IL phase and with ~5 mol% TBP added at 308 K and three pressures.

| Pressure (MPa) | Solubility (g Pr/L) | | |
|---|---|---|---|
| | PrTf$_2$N | Pr(hfacac)$_3$ | Pr(NO$_3$)$_3$ |
| 14 | 0.01 | 0.40 | <0.01 |
| 28 | 0.09 | 0.53 | <0.01 |
| 34 | 0.18 | 0.58 | 0.02 |

Based on the information in the tables, the fluorinated Pr salts were soluble in scCO$_2$ alone, without any IL or TBP additive. Indeed, metal salts like Pr(NO$_3$)$_3$, without fluorine or additives, are known to be largely insoluble in scCO$_2$.

However, when the Pr salts were dissolved in the BMPyrTf2N ionic liquid obtaining a 40 mM Pr solution, before carbon dioxide was added to create a two-phase IL/scCO$_2$ system, there was essentially no detectable amount of Pr in the scCO$_2$ phase, except perhaps in the extreme case with Pr(hfacac)$_3$ at 3.4 MPa. In these experiments, 0.7 mL of the 40 mM Pr solution was placed in the scCO$_2$ cell. At this concentration, this corresponded to $2.8 \times 10^{-5}$ moles Pr ($3.9 \times 10^{-3}$ g Pr). The volume of the scCO$_2$ cell was 3.5 mL, and assuming the 0.7 mL of the pressurized IL solution occupied about 0.5 mL, then the scCO$_2$ phase would occupy about 3 mL of the cell. If all the Pr was transferred from the IL phase to the scCO$_2$ phase, the concentration or Pr in the scCO$_2$ phase would have been $3.9 \times 10^{-3}$ gPr/3.0 mL or 1.3 gPr/L, well above the detection limit of 0.01 gPr/L. Thus, even if only 0.8% of the Pr salt moved from the IL phase into the scCO$_2$ phase, it would have been detected.

The effect of the TBP additive is very apparent from a comparison of the tabulated results in Tables 2 and 3. Without the additive, the amount of Pr that moved into the scCO$_2$ phase was essentially below the detection limits, but in the presence of TBP at ~5 mol%

concentration, based on the moles of $CO_2$ in the cell, the metal salts moved into the $scCO_2$ phase. The $Pr(hfacac)_3$ salt was exchanged most readily. From Table 3, the highest value of Pr in the $scCO_2$ phase was almost 0.58 g/L at 3.4 MPa of pressure. Assuming $scCO_2$ occupied about 2.5 mL of the cell volume after adding 0.7 mL of TBP, the amount of Pr in the $scCO_2$ phase was approximately 0.58 g/L × 0.0025 L or $1.4 \times 10^{-3}$ g Pr. For the $Pr(Tf2N)_3$ metal salt, the exchanged Pr resulted in a concentration of 0.18 g/L or $4 \times 10^{-4}$ g Pr. The amount of Pr initially placed in the ionic liquid was ~ $4 \times 10^{-3}$ g Pr. Thus, the fraction of Pr exchanged between the IL and the $scCO_2$ phase was approximately 1/3 and 1/10 at 3.4 MPa. Note that the amount exchanged between phases increased with an increasing pressure. In the case of the $Pr(NO)_3$ salt, the TBP additive had little effect on moving and solubilizing the salt in the $scCO_2$ phase.

To determine the amount of Pr that was distributed between the IL phase and the $scCO_2$ phase, the distribution coefficient, $K_d$, can be calculated as follows:

$$Kd = g \text{ of Pr in scCO}_2 \text{ phase}/g \text{ of Pr in IL phase}$$

When Pr was exclusively in the IL phase, $K_d$ reached its minimum value of 0. The distribution coefficient was calculated both with and without the tributylphosphate additive. Since in almost all cases, in the absence of the TBP additive, the amount of Pr present in the $scCO_2$ phase was below the detection limits, the distribution coefficient was zero or not calculable. Without TBP, Pr solubility was below detection in one or both phases. With the TBP additive, in most cases, the Pr content in $scCO_2$ was measurable. The values of the distribution coefficient are provided in Table 4, at 308 K, for all pressures investigated with the TBP additive.

**Table 4.** Values of the Distribution Coefficient for the $scCO_2$ phase with the TBP additive at 308 K.

| Pressure (MPa) | $K_d$ with TBP Additive | | |
| --- | --- | --- | --- |
| | $PrTf_2N$ | $Pr(hfacac)_3$ | $Pr(NO_3)_3$ |
| 14 | 0.005 | 0.34 | * NC |
| 28 | 0.059 | 0.50 | * NC |
| 34 | 0.13 | 0.56 | 0.01 |

* (NC, not calculable; Pr absorbance was below detection limits in one or both phases).

## 3. Discussion

The solubilities of several Pr salts in BMPyrTf2N IL were determined at room temperature. BMPyTf2N is a stable, hydrophobic IL with a large electrochemical window that has been used as a medium for the electrochemical deposition of REMs. This information may be useful to identify a Pr salt form for recovery through an electrochemical deposition process using the BMPyrTf2N ionic liquid. In all cases studied, the dried salts had higher solubility. Only three of the Pr salts exhibited a solubility greater than 40 mM in this IL, i.e., $Pr(NO_3)_3$, $Pr(Tf2N)_3$, and $Pr(hfacac)_3$.

The solubilities of these Pr salts were further investigated for the possible electrodeposition of Pr from a BMPyrTf2N IL/$scCO_2$ two-phase system. The fluorinated Pr salt compounds were found to be soluble in pure $scCO_2$, whereas $Pr(NO_3)_3$ had little or no solubility. The solubilities increased with an increasing pressure. If the metal was introduced in the $scCO_2$ cell as a 40 mM solution in the IL and then pressurized with $CO_2$ to form the IL/$scCO_2$ two-phase system, the Pr salt was not exchanged between the two phases. However, if tributylphosphate was added to the IL Pr salt solution, then some of the Pr salt moved into the supercritical $CO_2$ phase, and its solubility increased with an increasing pressure. Distribution coefficients, for the $scCO_2$ phase, with values as high as 0.56 and 0.13 were calculated for the $Pr(hfaca)_3$ and the $Pr(Tf2N)_3$ salts, respectively.

The fact that little to none of the Pr salt moved from the IL phase to the $scCO_2$ phase in the absence of any additive bodes well for performing the reduction of the metal

salt in a BMPyrTf2N IL/scCO$_2$ two-phase system. Conceivably this would allow for electrodeposition in the IL phase, while at the same time, oxidation products soluble in the scCO$_2$ would be transferred to the scCO$_2$ phase and removed. If the process called for removing some of the Pr salt from the IL phase, to perhaps purify it, this would be possible as well, and the amount moved into the scCO$_2$ phase could perhaps be tuned by adding the appropriate amount of additive. A specially designed cell with electrodes immersed just in the IL phase would be needed. If there was also a slow flow of the pressurized CO$_2$, then CO$_2$ could potentially be scrubbed of any impurities or oxidation products that might move into and collect inside the scCO$_2$ phase.

## 4. Materials and Methods

The solubility of several Pr(III) salts in 1-butyl-1-methylpyrrolidimium bis(trifluorome-thylsulfonyl)imide BMPyrTf2N were determined by preparing saturated solutions of the dried praseodymium salts in BMPyrTf2N. The praseodymium(III) salts used for the solubility studies were chloride (PrCl$_3$), carbonate (Pr$_2$(CO$_3$)$_3$), triflate (PrTf$_3$), acetate (Pr(C$_2$H$_3$O$_2$)$_3$), Tris(6,6,7,7,8,8,8-heptafluoro-2,2-dimethyl-3,5-octanedionato) (Pr(fod)$_3$), nitrate (Pr(NO$_3$)$_3$), acetylacetonate, Pr(acac)$_3$, hexafluoroacetylacetonate (Pr(hfacac)$_3$), and bis(trifluoromethylsulfonylimide) (Pr(Tf2N)$_3$). All Pr salts were obtained from commercial sources, except the Pr(Tf2N)$_3$ compound. Information regarding the commercially obtained Pr compounds used in the solubility studies is provided in Table 5.

**Table 5.** List of Praseodymium salts with the names of their suppliers and their purity.

| Pr Compound | Supplier | Purity |
|---|---|---|
| PrCl$_3$ | Sigma Aldrich, St. Louis, MO, USA | 99.9% |
| Pr$_2$(CO$_3$)$_3$ | Fisher Scientific, Waltham, MA, USA | 99.9% |
| PrTf$_3$ | Fisher Scientific | 98% |
| Pr(C$_2$H$_3$O$_2$)$_3$ | Alfa Aesar, Tewksbury, MA, USA | 99.9% |
| Pr(fod)$_3$ | TCI America, Portland, OR, USA | >97% |
| Pr(NO$_3$)$_3$ | Sigma Aldrich | 99.9% |
| Pr(acac)$_3$ | Sigma Aldrich | 99.9% |
| Pr(hfacac)$_3$ | Sigma Aldrich | >97% |
| Pr(Tf2N)$_3$ | synthesized in lab | - |

The Pr(Tf2N)$_3$ compound was synthesized in the laboratory through anion exchange with the carbonate salt, following the procedure used previously by other researchers to synthesize Dy(Tf2N)$_3$ [19].

The salts were dried by placing them in a vacuum oven for a minimum of 4–5 h at a temperature of 90 °C and a vacuum of ~2 kPa. These conditions were deemed sufficient, since the weight of the dried material after 4 h was essentially constant after placing it back in the vacuum oven for another half hour to hour. The dried salts were placed in a glove box, and approximately 0.5 g of the salt was added to 1.0 mL of the BMPyrTf2N IL. The solution was placed in a vial with a small magnetic stir bar, sealed, and then stirred for 6–8 h at an elevated temperature of ~35 °C. In most cases, this amount of salt was sufficient to create a saturated solution. If the salt dissolved completely, then additional salt was added, and the process was repeated until some salt remained.

To prepare a saturated solution for the determination of Pr concentration by absorbance measurements, the saturated Pr salt solution was cooled to room temperature (23 °C) and pushed through a 0.5-micron filter using a syringe to remove any solid that might scatter light and distort the absorbance measurements. The absorbance values of the solutions were measured with a Shimadzu Model UV-3101PC UV–Vis spectrometer using a 1 cm pathlength reduced-volume quartz cuvette. The solutions were diluted with methanol to

keep the solution absorbance below 1. As there was a wide range of the solubilities of the Pr salts, the amount of methanol added varied. The concentrations of Pr in the saturated solutions were determined from the Beer–Lambert -aw based on the molar absorptivity value, $\varepsilon$, at $\lambda_{max}$ of the Pr(III) absorption peak at ~590 nm and the absorbance value of the saturated metal salt solution at $\lambda_{max}$, taking into account the dilution factor due to the added methanol. The molar absorptivity, $\varepsilon$, at $\lambda_{max}$ was found by measuring the absorbance value at $\lambda_{max}$ for a solution containing Pr(III) at a known concentration. The measured $\varepsilon$ value of 1.8 L/(mol cm) was used in the Beer–Lambert law, A = $\varepsilon$ c l to find the concentration of Pr(III) in each saturated salt solution.

Measurements of the absorbance values at $\lambda_{max}$ were also used to measure the solubility of Pr compounds in $scCO_2$. Since the critical temp. and pressure were 304.13 K and 7.377 MPa, a special high pressure $scCO_2$ cell was constructed, with 25 mm-diameter sapphire windows for in situ spectroscopic measurements. The cell contained special insert spaces into which cylindrical heaters could be inserted for temperature control. An Omega temperature control unit was used to control the temperature of the $scCO_2$ cell. The supercritical cell was described previously [20]. The volume of the inside of the cell was 3.5 $cm^3$, and the pathlength of the cell was 1.5 cm.

Solubility measurements of selected Pr salts were made in neat $scCO_2$ and also in the $scCO_2$ phase of the BMPYrTf2N IL/$scCO_2$ two-phase system. There are reasons for determining the Pr salt solubility in neat $scCO_2$ and also in the $scCO_2$ phase of the IL/$scCO_2$ 2-phase system when the salt is first dissolved in the IL. If the $scCO_2$ is to be used to remove impurities and oxidation products during electrochemical deposition, then even if the Pr salt is soluble in $scCO_2$, ideally once the Pr salt is loaded into the IL, it will remain completely in the IL phase of the system without being transferred into the $scCO_2$ phase. If it does not transfer to the $scCO_2$ phase, then it cannot be carried out with the impurities. In another scenario, it might be advantageous to transfer a controlled amount of the metal salt into the $scCO_2$ to purify it, if the oxidation products or impurities become tethered onto the Pr salt. Tributylphosphate, TBP, is known to be a good metal extractant, and measurements were also made to determine how its presence in the ionic liquid affected the amount of metal salt transferred from the IL phase into the $scCO_2$ phase. These solubility measurements were made for only the salts which showed solubilities greater than 5 g Pr/L soln. (~35 mM Pr) in the BMPyTf2N ionic liquid. At 5 g Pr/L, it seems reasonable that the use of electrodeposition for recovery of Pr metal might be marginally cost-effective. Ionic liquids have been used for electroplating Al in an industrial process [21]. Compounds with higher solubility would theoretically provide a much better yield.

Generally speaking, metal salts are known to have limited solubility in $scCO_2$, but the presence of fluorine often causes a significant change in their solubility. The procedure used to measure the solubility of the selected metal salts involved first adding a small amount of solid to the $scCO_2$ cell and then bringing the $CO_2$ in the supercritical fluid cell to the proper desired temperature and pressure. A temperature of 308 K and pressures of 14, 28, and 34 MPa of $CO_2$ were investigated. The cell was placed in a specialized holder constructed at the Idaho National Lab, and once the cell reached the correct temperature and pressure, the absorbance of the contents of the $scCO_2$ cell at 590 nm was measured through the sapphire windows. The instrument was first zeroed using the empty cell with its windows attached.

For the measurements of Pr solubility in the $scCO_2$ phase of the two-phase BMPyrTf2N IL/$scCO_2$ system, a 40 mM solution of the Pr salt was prepared, 0.7 mL of it was placed into the $scCO_2$ cell, and then the carbon dioxide was added to the cell. A magnetic stir bar stirred the contents, and after the cell reached 308 K and the desired pressure, the absorbance of the peak at 590 nm was measured. Pressure of 14, 28, and 34 MPa were studied here as well. Upon completion of these experiments, 0.7 mL of TBP was added to the Pr salt IL solution in the cell, and the absorbance at the 590 nm peak was remeasured. This amount of TBP was roughly 5 mol%, based on the moles of $CO_2$ present at a pressure

of 30 MPa. Care was taken to ensure that the absorbance measured was from the scCO$_2$ phase in the cell.

**Author Contributions:** All authors contributed to the work realized in this article in the following ways: (1) conceptualization, R.F., D.B. and R.R.; (2) methodology, R.R., D.B. and M.C.; (3) resources R.F. and D.B.; (4) original draft preparation, R.R.; writing—review and editing, R.R., D.B. and R.F.; funding acquisition, D.B. and R.F. All authors have read and agreed to the published version of the manuscript.

**Funding:** This research was funded by the INL Laboratory Directed Research & Development (LDRD) Program under DOE Idaho Operations Office Contract DE-AC07-05ID14517.

**Data Availability Statement:** Please contact the corresponding author for details regarding data availability.

**Acknowledgments:** Rodriguez would like to acknowledge sabbatical support received from Idaho State University.

**Conflicts of Interest:** The authors declare no conflict of interest. The funders had no role in the design of the study; in the collection, analyses, or interpretation of data; in the writing of the manuscript, or in the decision to publish the results.

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
