# Peer review of "Studies toward the Use of Ionic Liquids and Supercritical CO2 for the Recovery and Separation of Praseodymium from Waste Streams"

_catalysts, doi:10.3390/catal12030335_

Round 1

Reviewer 1 Report

In this manuscript, the authors studied the solubility of various Pr compounds in BMPyTf2N ionic liquid and their distribution in supercritical CO2 phase of the two phase BMPyTf2N ionic liquid/supercritical CO2 system. They have found that little to none of the Pr salt moves from the IL phase to the supercritical CO2 phase in the absence of any additive, which allows for electrodeposition in the ionic phase while the oxidation products soluble in the supercritical CO2 are transferred and removed. These results may help the recovering of the Pr element from incineration of metal containing materials. Overall, this is a well-written paper, and I recommend its publication. 

Author Response

Reviewer did not have any comments that needed to be addressed.

Reviewer 2 Report

The article „Studies Toward the Use of Ionic Liquids and Supercritical CO2 2 for the Recovery and Separation of Praseodymium from Waste Streams“ of authors „Rene Rodriguez, Donna Baek, Mary Case and Robert Fox“ deals with extraction of praseodymium salts into the selected ionic liquids and with possibility to utilize scCO2 for impurities separation. The article compares solubility of different Pr(III) salts with inorganic and organic anions in 1-butyl-1-methylpyrrolidinium bis(trifluoromethylsulfonyl)imide and in scCOor mixture of scCO2/tributyl phosphate. The solubility was measured by absorbance measurements applying Beer-Lambert law.

This research is very useful, however, is it in the area of scope of Catalysts journal? Please, explain your choise of this journal.

Comments:

  1. Please, reconsider the title and text of abstract. Your article deals with solubilities measurements, not with waste streams treatment.
  2. Please, explain the abbreviations used on page 4, in Table 1 (hfacac, C2H3O2, FOD).
  3. On page 2, line 71 is abbrev. TfO- applied for CF3SO3- anion, however, in Table 1, formula PrTf3. Did you mean Pr(III) triflate (Pr(OTf)3 or not?
  4. Please check carefully all typo mistakes (page 3, line 125; Figure 2, formula of Pr(III) carbonates; page 4, lines 154-155; etc.)

Round 2

Reviewer 2 Report

The manuscript is corrected efficiently and ready for publishing.